# Regional β-Diversity of Stream Insects in Coastal Alabama Is Correlated with Stream Conditions, Not Distance among Sites

**DOI:** 10.3390/insects14110847

**Published:** 2023-10-31

**Authors:** Carlos Sankone, Chris Bedwell, John McCreadie

**Affiliations:** 1Biology Department, University of South Alabama, Mobile, AL 36688, USA; cms1822@jagmail.southalabama.edu; 2Bedwell Biological LLC., 2617 Grey Stone Rd, Henderson, NV 89074, USA; chrisrbedwell@gmail.com

**Keywords:** diversity, aquatic insects, streams, Alabama, turnover, nestedness, coastal

## Abstract

**Simple Summary:**

Biodiversity is measured differently depending on the spatial scale of the study. Beta (β)-diversity, for example, is a calculated measurement that addresses changes in diversity across different assemblages in a specific region. Studies of β-diversity often address changes in diversity across gradients of environmental conditions and distance among sites. However, β-diversity of stream insects can be difficult to measure due to their hyper-diversity and challenging taxonomy. Our study investigated the association of β-diversity with habitat conditions and distance among these habitats for insects found in the coastal streams of Alabama, USA. Additionally, we looked for potential influences caused by seasonality (fall and summer) and the level of taxonomic identification (genus, species). Regardless of season, stream conditions were highly correlated with β-diversity. More specifically, stream size and water chemistry showed the highest associations with β-diversity. Changes in β-diversity were largely driven by species replacement (turnover) rather than species loss (nestedness). The taxonomic resolution had minimal effects on the calculations of β-diversity across environmental conditions. Distance among stream sites was never correlated with β-diversity. As we continue to face global insect declines, our study provides valuable insight into the patterns that drive changes in diversity across environmental gradients.

**Abstract:**

β-diversity is often measured over both spatial and temporal gradients of elevation, latitude, and environmental conditions. It is of particular interest to ecologists, as it provides opportunities to test and infer potential causal mechanisms determining local species assemblages. However, studies of invertebrate β-diversity, especially aquatic insects, have lagged far behind other biota. Using partial Mantel tests, we explored the associations between β-diversity of insects found in the coastal streams of Alabama, USA, and stream conditions and distances among sites. β-diversity was expressed using the Sørensen index, *β_Sor_*, stream conditions were expressed as principal components (PCs), and distances as Euclidean distances (km) among sites. We also investigated the impact of seasonality (fall, summer) and taxonomic resolution (genus, species) on *β_Sor_*. Regardless of season, *β_Sor_* was significantly correlated (*p* < 0.01; r > 0.44) with stream conditions (stream size and water chemistry), while taxonomic resolution had minimal effect on associations between *β_Sor_* and stream conditions. Distance was never correlated with changes in *β_Sor_* (*p* > 0.05). We extended the use of the Sørensen pair-wise index to a multiple-site dissimilarity, *β_Mult_*, which was partitioned into patterns of spatial turnover (*β_Turn_*) and nestedness (*β_Nest_*). Changes in *β_Mult_* were driven mostly by turnover rather than nestedness.

## 1. Introduction

A fundamental pattern of great interest to ecologists is how and why species diversity changes with location or time. Species diversity is scale dependent [1,2] and, as such, is generally examined with regards to three spatial levels: alpha (α)-, beta (β)-, and gamma (γ)-diversity [3,4,5]. Alpha diversity is the number of species in a local assemblage, whereas γ-diversity is the number of species in a regional pool. The change in species composition among local assemblages within a region is β-diversity. One can view β-diversity as the measurement that links α- and γ-diversity within a region [4,6]. β-diversity is of particular interest to ecologists as it provides an opportunity to test and infer potential causal mechanisms determining local species assemblages [7,8].

Although β-diversity is simple in concept, there are a bewildering number of indices, with no clear consensus for meaning, measurement, statistical analysis, or interpretation [8,9,10,11,12]. One reason for multiple perspectives is that β-diversity is a calculated index as opposed to an observed or estimated value, such as α-diversity and γ-diversity. However, two broad approaches have emerged. The first approach examines directional changes in species composition across local assemblages over measured or inferred gradients, such as distance or local environmental conditions [8,12]. The second approach examines changes across local assemblages without regard to a specific gradient and thus can be considered a measure of variation in species composition [12]. This second approach was expressed by Whittaker’s measurement, β = γ ÷ α, in which β-diversity is the ratio of γ-diversity and mean α-diversity [12]. We explore the former approach by examining directional changes in β-diversity over gradients of stream conditions and distance among sites.

Studies have examined patterns of β-diversity over various gradients, such as altitude [13,14], environmental conditions [15,16], and latitude [17,18,19]. However, studies of invertebrate β-diversity [20], especially stream insects, lag far behind studies of other biota [14]. Such studies are particularly important in stream systems, where the high biodiversity of organisms in the low area of the streambed results in habitats that are especially vulnerable to global changes [21]. Accordingly, these habitats are some of the most endangered ecosystems on the planet [22].

Though some studies have been conducted in similar stream systems in the southern United States [23,24] little is known about the ecology of the coastal streams of Alabama, which ultimately flow into the Gulf of Mexico [25,26]. These streams tend to be short, usually less than 40 km, with few, if any, tributaries. These sandy-bottom streams are generally meandering and slow-moving, flowing mostly through forested habitat. Like other freshwater systems, these streams are especially vulnerable to global changes [21], which makes them a sensitive “biological meter” for measuring widespread changes in biodiversity. These streams are of particular interest as many drain into what E.O. Wilson referred to as “America’s Amazon”—the Mobile Tensaw Delta (MTD). The MTD is a 1150-km^2^ relatively pristine area consisting largely of cypress-gum swamps and bottomland hardwood forests interlaced with large rivers, streams, canals, bayous, and marshes [27].

We calculated β-diversity as Sørensen pair-wise indices, *β_Sor_*, among pairs of stream sites [28]. *β_Sor_* is calculated using presence/absence data and can be expressed as:$$\beta_{Sor}=\frac{2C}{S1+S2}$$
where *C* is the number of species the two communities have in common, *S*1 is the total number of species found in community 1, and *S*2 is the total number of species found in community 2 [11]. Hence, this index can be viewed as the dissimilarity in species composition between any two sites.

Accordingly, our aim was to explore how patterns of *β_Sor_* of lotic insects in these historically understudied coastal streams of Alabama changed over gradients of environmental conditions and distance. We also explored how seasons (fall and summer) and different taxonomic resolutions (genus only, genus/species mix, and species only) influenced patterns of *β_Sor_*. We then extended the use of the Sørensen pair-wise index to a multiple-site dissimilarity index, *β_Mult_*. [29]. Whereas Sørensen is an index of two sites in a region, *β_Mult_* provides a measurement of dissimilarity among all sites in a region. This latter index can further divide into diversity due to species turnover (change) among sites (*β_Turn_*) and species nestedness (loss) among sites (*β_Turn_*). Hence, *β_Mult_ = β_Turn_ + β_Nest_*.

## 2. Materials and Methods

### 2.1. Study Sites and Sampling Protocols

Details of the study area, sampling protocols, and measurements of stream conditions can be found in McCreadie and Bedwell [25]; here we present a brief summary for clarity. The study area is within the Southern Pine Hills and Coastal Lowlands districts of the lower Coastal Plain of Alabama, USA [30]. Watersheds in this area drain into the Mobile-Tensaw Delta, Mobile Bay, or the Gulf of Mexico. These coastal streams are low-gradient, meandering systems, with sand as the primary streambed substrate. Twenty-three streams between 30.74–31.25° N and 87.76–88.24° W (Figure 1) were sampled from 2 October to 17 November 2007 (fall collections), with 17 of these streams sampled from 29 May to 29 June 2007 (summer collections). Mean Euclidean distance (±SE) among sites in the fall was 26.7 ± 0.67 km and for summer sites, 25.6 ± 0.96 km.

At each stream site, a 100-m representative length of stream (scale = local assemblage) was delineated, and samples (*n* = 5), taken (randomly) within this length at each site for both the fall and summer sampling regimes. Insects were sampled using a standard 1 m × 1 m kick-net, mesh size 0.3 mm, over a standard quadrate size (PVC pipe) of 60 cm × 60 cm [31]. The streambed was dominated by shifting sand, with insects confined to isolated habitat patches such as snags or leaf packs. Using the protocols of McCreadie et al. [32], the following stream variables were recorded at each collection: width (W), depth (D), velocity (U), conductivity, pH, temperature, dissolved oxygen, riparian vegetation, canopy cover, and dominant streambed particle size. Riparian vegetation (open, brush, and forest), canopy cover (none, partial, and complete), and streambed (mud, sand, gravel, and cobble) were measured as rank variables. Discharge, Q (m^3^ s^−1^), was calculated as “W × D × U = Q”. Latitude and longitude were recorded using Garmin GPS units (Rino 110) (Kansas City, MO, USA). Water hardness was measured using CHEMetric^®^ (Midland, VA, USA) titration cells. The above variables are considered to be influential determinants (or at least strong predictors) of species assemblages in stream habitats [33]. Range and mean stream conditions by site for both summer and fall collections are given in Appendix A.

Samples were transported to the laboratory on ice and preserved in 95% ethanol until identification. For each sample, an effort was made to remove all insects, with no samples being sub-sampled. Of the separated insects, five taxonomic orders were selected for study due to their high abundance, functionalities, ease of identification, and broad representation of functional feeding groups. Specimens were identified following the keys of Epler [34] for Coleoptera, Pescador et al. [35] for Plecoptera, Rasmussen and Pescador [36] for Megaloptera, Daigle [37] and Richardson [38] for Odonata, and Pescador and Rasmussen [39] for Trichoptera. Ideally, identification should be at the species level; however, larvae of many aquatic species can only be identified by genus (e.g., *Leuctra*). In addition, while species keys exist, they are often only useful for later instars. Accordingly, all specimens were identified by either the level of genus or species. Hence, three taxonomic levels were subject to statistical analysis—mixed taxa of genus/species, generic data only, and species data only. Voucher specimens for identification were deposited in the University of South Alabama’s Arthropod Depository.

### 2.2. Statistical Analyses

All statistical tests were considered significant at *p* < 0.05, unless otherwise stated. To determine if our sampling protocol produced a reasonable estimate of the species present in the regional pool, species accumulation curves, using a bootstrap approach [40], were calculated for each combination of season (fall, summer) and level of identification (mixed genus/species, genus only, species only), with six calculations produced in total. Bootstrap estimators were based on 999 random sample-based re-orderings of observed taxonomic grouping for each level of sampling effort (number of sites), with 1 to 23 reorderings for fall collections and 1 to 17 for summer data. The maximum bootstrap estimator of species richness was then compared to the observed species richness for each season/taxonomic level combination.

Each partial Mantel test [41] was based on three matrices constructed for each season/taxonomic level combination. These matrices included one for distance, one for stream conditions, and one for β-diversity. Distance was expressed as the Euclidean pairwise distance between each site (km). Stream conditions were log-transformed and subsequently subjected to principal component analyses (PCA) for each season. PCs with eigenvalues greater than 1.00 were selected to represent stream variable proxies [42]. The interpretation of each PC was based on correlations between each PC and the original stream variable (*p*-value set at *p* < 0.01) to provide a rigorous interpretation of the PCs [42]. The third matrix was the pairwise comparison of the Sørensen index, *β_Sor_*, a comparison of diversity between each site [12].

For each set of matrices, partial Mantel tests [41] were used to detect significant associations between the *β_Sor_* andstream conditions (accounting for distance effects) and the *β_Sor_* and distance matrix (accounting for stream conditions). Partial Mantel tests were calculated separately for both season and taxonomic resolution, with the significance of each partial correlation based on 5000 random permutations between matrices. Mantel tests have been used by previous authors and are considered a powerful approach for detecting spatial correlations [43].

We then extended the use of the *β_Sor_* pairwise index to a multiple-site dissimilarity index, *β_Mult_*, following Baselga [44]. *β_Mult_* can further partition the pattern of species occurrences in the site-by-species matrix into spatial turnover (*β_Turn_*) and nestedness (*β_Nest_*) or combinations thereof [44,45]. These two broad processes—replacement of species (spatial turnover) and species loss (nestedness)—determine the pattern of species occurrences on the site by species matrix [43]. These were presented as single measurements for each combination of season x taxonomic level data sets. Nestedness among local assemblages results when assemblages with fewer species are subsets of species in larger assemblages, reflecting processes that result in species loss from local assemblages [44,46,47]. Spatial turnover results from the replacement of species with other species across local assemblages [44].

PCA was conducted using Minitab v 20, and bootstrap estimators of number of taxa using Primer v 6.0 [48]. Partial Mantel tests were conducted using the PAST software package v. 190 [49], and multiple-site measures of diversity were calculated using the R package Betapart v 1.5.1. [29].

## 3. Results

### 3.1. Collection and Identification

Table 1 summarizes the genera found in over 50% of sites, with the entire fauna collected presented in Appendix A. For the summer collections, a total of 6281 specimens were identified from 17 stream sites, belonging to either genus or species, in 27 families and 61 genera. The most common genera encountered (>80% of sites) included the Elmidae (*Dubiraphia*, *Stenelmis*) and Leuctridae (*Leuctra*). A total of 20,739 specimens were identified from 23 sites in the fall for either genus or species, in 27 families and 66 genera. In these collections, the most frequent genera encountered (>80% of sites) included the Elmidae (*Ancyronyx*, *Stenelmis*), Coenagrionidae (*Argia*), Philopotamidae (*Chimarra*), Hydropsychidae (*Hydropsyche*, *Cheumatopsychidae*), Gomphidae (*Progomphus*), Perlidae (*Acroneuria*), and Leuctridae (*Leuctra*).

Differences between the number of taxa observed and the estimated bootstrap estimate of taxa (Table 2) were between 7.3 and 11.6%, with all estimated taxa higher than the observed number of taxa. We interpreted this to indicate a reasonable representation of the local insect stream fauna during the dates of collection.

### 3.2. Principal Component Analyses of Stream Variables

Five PCs were selected to express stream conditions for both the fall and summer data sets. Correlation analysis (*p* < 0.001) for the fall collections (Table 3) suggested PCs were largely a measure of stream size (discharge, depth) and water column variables (pH, conductivity, temperature, and hardness). In a similar vein, correlation analysis (*p* < 0.001) for the summer collections (Table 4) indicated PCs were also a measure of stream size (width and depth) and water column variables (pH, conductivity, oxygen, and hardness), in addition to the type of riparian vegetation (i.e., ranked as open, brush, or forest) [32].

### 3.3. Partial Mantel Test Correlations

Results of the partial Mantel tests (Table 5) showed that changes in *β_Sor_* among sites were highly correlated (*p* = 0.0041–0.0001) with stream variables expressed as PCs. Given the correlations between stream variables and PCs, we interpreted this to indicate that β-diversity changed largely along gradients of stream size and water chemistry for both fall and summer collections, as well as riparian vegetation for summer collections. Mantel test correlations for species-level identification data were slightly lower than data for generic or mixed levels, regardless of season (Table 5). In contrast, distance among sites was never correlated with β-diversity (Table 5).

*β_Mult_* was relatively consistent among all combinations of season and level of identification (*β_Mult_* = 0.8210–0.8616). In addition, changes in *β_Mult_* diversity among sites were largely the result of species turnover (*β_Turn_* = 0.7084 to 0.7916) as opposed to local species loss (*β_Nest_* = 0.0697–0.1220) (Table 5).

## 4. Discussion

We examined two aspects of β-diversity: *β_Sor_* was calculated as the dissimilarity for each pairwise site comparison, whereas *β_Mult_* extended to all sites, producing a single estimate of overall β-diversity. Our results suggest that stream conditions had a profound effect on β-diversity of lotic insects, as indicated by the significant correlations between site conditions (expressed as PCs) and *β_Sor_*. In contrast, spatial factors (i.e., distance among sites) showed no significant correlations with *β_Sor_* at a regional scale. The level of taxonomic precision (species, mixed, and genus) had minimal influence on the correlation, with *β_Sor_* correlations using species-level identification being marginally lower than data using mixed or generic-level identifications. Correlations between stream condition and taxonomic level of identification showed little difference between seasons (fall and summer). The complementary analysis of multisite estimates of β-diversity indicated that species turnover, *β_Turn_*, was consistently higher than species loss, *β_Nest_*, for all data sets.

β-diversity is a measurement of differences in species composition among sites across gradients of habitat conditions or time. Accordingly, there has been substantial interest in β-diversity recently [8,11,12,44,50] as a means of understanding the causal factors that both determine and modulate community assemblages over space and time. In the current study, as stream conditions diverged among sites, so did the composition of the lotic insect assemblages. Changing habitat conditions over a gradient increases the number of niches and the potential for more species; thus, variation in species composition should increase among habitats [51,52], which in turn should increase β-diversity, as seen in our study. Correlation analysis with PCs and stream conditions suggested that the stream size and water column conditions were the main drivers of species replacement among sites. Stream size and water column conditions have been shown to be important factors (or at least predictors) in shaping local species assemblages of stream macroinvertebrates in both tropical and temperate streams [50,53,54,55,56].

Change in β-diversity among habitats is generally considered the result of two different processes, species turnover and species loss or gain [43], measured by *β_Turn_* and *β_Nest_*, respectively. Species turnover is generally viewed as a result of niche-based species sorting along abiotic and biotic factors [43,44,57], although other processes can determine local assemblages [57,58]. Our analysis of partitioning *β_Mult_* into these components of turnover and nestedness shows species turnover (*β_Turn_*) was much higher than nested patterns (*β_Nest_*) in all six data sets. Hence, *β_Turn_* is essentially a measure of *β_Mult_* at a regional scale in our study. To date, the majority of studies that have partitioned *β_Mult_* show that changes in species composition are largely the result of species turnover (*β_Turn_*), with nested effects having minimal influence at regional scales [59]. Nestedness has often been viewed as a limitation in dispersal ability, particularly in aquatic habitats [60,61]. The consistently low *β_Nest_* values and the nonsignificant results of distance in the Mantel tests suggest dispersal limitation was not an important mechanism for structuring species assemblages. However, the use of spatial variables as proxies for dispersal is not without criticism [62]. The low influence of spatial factors on insects at a regional level may be the result of at least three factors. First, all the insects we investigated have a winged adult stage, which would promote dispersal ability [50]. The second factor is a distribution/time element. Hence, though a species may not be able to directly disperse between two large distances, intervening habitats (i.e., other streams) could provide a means of ‘leap frogging’ to new sites. Given enough time, a species could then disperse to all viable sites within a region. It must also be noted that the scale of our study is relatively small (mean distance between sites ~25–30 km), which may have limited the influence of distance.

The minimal influence of taxonomic resolution (genus, mixed, and species) may be advantageous for studies of diversity that include taxa that are difficult or impossible to identify as species based on the morphology of early instars. We suggest the minimal effect of the season (fall, spring) in our results may be due to the fact that many of the species encountered in our study are multivoltine and are thus present throughout most or all of the year.

## 5. Conclusions

Despite the importance of aquatic insects as bioindicators of stream health, little is known about the spatiotemporal patterns of aquatic insect diversity. Additionally, even less is known about those that inhabit the coastal streams of Alabama, which drain into the highly diverse Mobile-Tensaw River Delta. Researching patterns of diversity in these insect communities has become especially pertinent as we continue to face a global crisis of insect decline. Our study provides foundational insight into the influences that drive aquatic insect diversity in coastal Alabama, but there is still more work to be done to understand the specific impacts and extent of these environmental changes over time. We suggest that future studies address the impacts of stream conditions on coastal Alabama’s insect diversity across both spatial and temporal scales.

## Figures and Tables

**Figure 1 insects-14-00847-f001:**
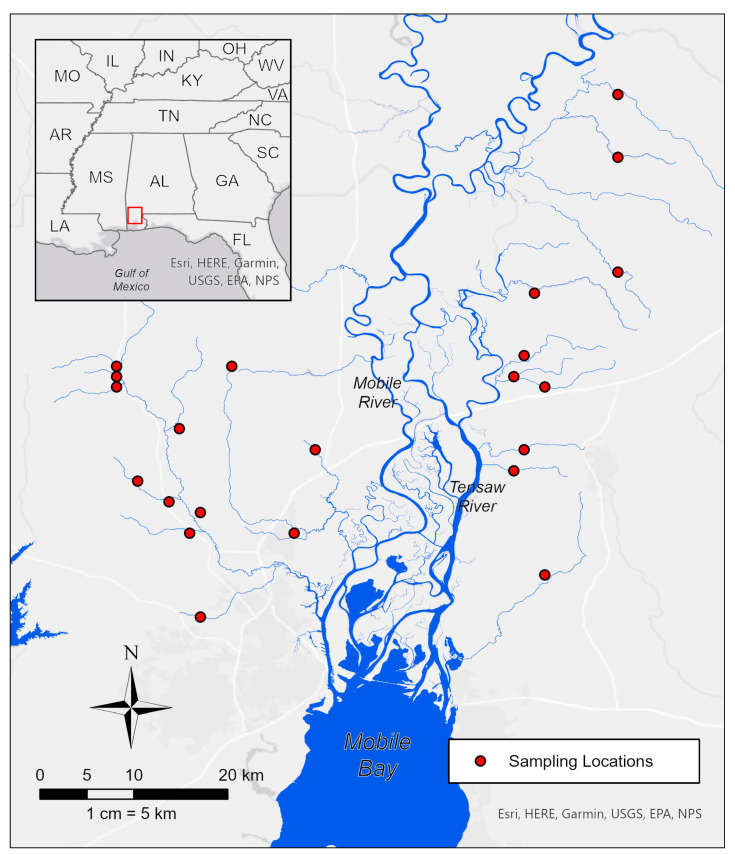
Location of stream sites sampled from October to November 2007, and May to June 2007. The inset map displayed in the top-left shows the proximity of the study region with respect to the Gulf of Mexico and surrounding states. The red outline within the inset map indicates the current map view.

**Table 1 insects-14-00847-t001:** Family and genera found in over 50% of sites for the summer (29 May to 29 June 2007) and fall (2 October to 17 November 2007) collections. A complete list of all genera is given in Appendix A.

Summer Collections			Fall Collections		
Family	Genus	% Occurrence	Family	Genus	% Occurrence
Elimidae	*Ancyronyx*	94.1	Elmidae	*Ancyronyx*	100.0
Aeshnidae	*Boyeria*	94.1	Elmidae	*Stenelmis*	95.7
Elimidae	*Stenelmis*	94.1	Coenagrionidae	*Argia*	91.3
Elimidae	*Dubiraphia*	82.4	Philopotamidae	*Chimarra*	91.3
Leuctridae	*Leuctra*	82.4	Hydropsychidae	*Hydropsyche*	91.3
Gomphidae	*Gomphus*	76.5	Gomphidae	*Progomphus*	91.3
Hydropsychidae	*Chimarra*	70.6	Hydropsychidae	*Cheumatopsyche*	87.0
Elimidae	*Gonielmis*	70.6	Perlidae	*Acroneuria*	82.6
Perlidae	*Acroneuria*	64.7	Leuctridae	*Leuctra*	82.6
Perlidae	*Neoperla*	64.7	Gomphidae	*Gomphus*	78.3
Libellulidae	*Neurocordulia*	64.7	Libellulidae	*Neurocordulia*	78.3
Hydropsychidae	*Cheumatopsyche*	58.8	Leptoceridae	*Oecetis*	78.3
Macromiidae	*Macromia*	58.8	Calamoceratidae	*Anisocentropus*	73.9
Gomphidae	*Progomphus*	58.8	Elmidae	*Gonielmis*	73.9
Perlidae	*Perlesta*	52.9	Elmidae	*Microcylloepus*	73.9
			Elmidae	*Dubiraphia*	69.6
			Perlidae	*Perlinella*	69.6
			Aeshnidae	*Boyeria*	65.2
			Brachycentridae	*Brachycentrus*	65.2
			Perlidae	*Paragnetina*	65.2
			Psephenidae	*Ectopria*	56.5
			Calopterygidae	*Calopteryx*	52.2
			Polycentropodidae	*Neuroclipsis*	52.2

**Table 2 insects-14-00847-t002:** Observed number of taxa for each data set and the bootstrap estimate of the number of taxa.

Data Set	Observed No. of Taxa	Bootstrap Estimator of Taxa
**Fall**		
Genus/species	98	107.8
Genus	75	80.9
Species	44	49.2
**Summer**		
Genus/species	83	93.9
Genus	62	67.8
Species	50	56.5

**Table 3 insects-14-00847-t003:** Principal component analysis of stream variables and correlation analysis of principal component scores to the original stream variables, fall 2007. Mean values and ranges of the stream variables are listed in Appendix A.

	Principal Components (PC)				
	PC1	PC2	PC3	PC4	PC5
**Eigen Analysis**					
Eigenvalue	3.1060	2.7122	1.9497	1.3903	1.1557
% Proportion variance explained	0.239	0.209	0.150	0.107	0.089
% Cumulative variance explained	0.239	0.448	0.598	0.704	0.793
**Correlation analysis ^1^**					
Depth	0.363	−0.650 **	0.377	0.343	−0.190
Velocity	0.622 *	0.184	−0.081	0.526 *	−0.240
Elevation	−0.449	−0.567 *	0.065	0.126	0.379
Width	0.511	0.215	0.276	−0.548 *	0.060
Discharge	0.815 **	−0.196	0.343	0.148	−0.153
Temperature	0.110	−0.252	0.372	−0.743 **	−0.211
pH	0.359	0.314	0.649 **	0.086	0.158
Dissolved O_2_	0.537 *	0.298	−0.426	0.016	−0.278
Conductivity	−0.409	0.662 **	0.500	0.133	0.134
Hardness	−0.240	0.707 **	0.541 *	0.139	−0.052
Canopy	−0.472	−0.454	0.457	−0.053	−0.359
Riparian Veg.	0.394	−0.586 *	0.216	0.135	0.506
Bed Substrate	0.558 *	0.382	−0.146	−0.199	0.477

^1^ *p* < 0.01 *, *p* < 0.001 **.

**Table 4 insects-14-00847-t004:** Principal component analysis of stream variables and correlation analysis of principal component score and original stream variables, summer 2007. Mean values and ranges of the stream variables are listed in Appendix A.

	Principal Components (PC)				
	PC1	PC2	PC3	PC4	PC5
**Eigen Analysis**					
Eigenvalue	3.5595	2.5328	1.9530	1.4662	1.1505
% Proportion variance explained	0.274	0.195	0.150	0.113	0.089
% Cumulative variance explained	0.274	0.469	0.619	0.732	0.820
**Correlation analysis ^1^**					
Depth	0.628 *	0.188	−0.535	0.026	0.175
Velocity	0.016	−0.621 *	−0.218	−0.091	−0.059
Elevation	0.551	0.318	−0.156	0.058	0.373
Width	0.176	−0.803 **	−0.330	−0.021	0.228
Discharge	0.590	−0.352	−0.669 *	0.050	0.043
Temperature	−0.530	−0.139	0.407	−0.196	0.578
pH	−0.323	−0.166	−0.071	−0.789 **	0.338
Dissolved O_2_	−0.125	−0.855 **	0.183	0.157	0.011
Conductivity	−0.777 **	−0.011	−0.487	−0.302	−0.115
Hardness	−0.720 **	0.227	−0.422	−0.248	−0.345
Canopy	0.269	0.626 *	−0.233	−0.224	0.289
Riparian Veg.	0.799 **	−0.122	0.322	−0.358	−0.262
Bed Substrate	0.483	−0.272	0.353	−0.606 *	−0.426

^1^ *p* < 0.01 *, *p* < 0.001 **.

**Table 5 insects-14-00847-t005:** Partial Mantel tests, matrix fill, and multisite β-diversity estimates ^1^.

Data Set	Partial Mantel Correlations ^2^		Matrix Fill ^3^	*β_Mult_*	*β_turn_*	*β_nest_*
	Stream Conditions	Distance among Sites				
**Fall**						
Genus/species	0.6693 *p* = 0.0001	−0.0612 *p* = 0.7100	0.1852	0.8471	0.7692	0.0778
Genus	0.6333 *p* = 0.0001	−0.1071 *p* = 0.8907	0.1417	0.8291	0.7297	0.0994
Species	0.4464 *p* = 0.0003	0.0684 *p* = 0.2098	0.0831	0.8616	0.7916	0.0697
**Summer**						
Genus/species	0.6182 *p* = 0.0013	0.1458 *p* = 0.1216	0.2871	0.8436	0.7525	0.0911
Genus	0.5859 *p* = 0.0028	0.1436 *p* = 0.1259	0.2145	0.8210	0.7084	0.1126
Species	0.4980 *p* = 0.0041	0.1776 *p* = 0.0619	0.1730	0.8479	0.7280	0.1200

^1^ *β_Mult_* = total β-diversity among all sites. *β_Turn_* = β-diversity due to species turnover. *β_Nest_* = β-diversity due to loss (nested pattern). ^2^ Top value partial correlation ^3^ Matrix fill is the proportion of cells in the species matrix with at least one taxon compared to the total number of cells in the matrix.

## Data Availability

Data is available (from JW McCreadie) to all interested parties once the manuscript is published. All generic identification of frequency of occurrence across sites is given in Appendix A.

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
