# Peer review of "Regional β-Diversity of Stream Insects in Coastal Alabama Is Correlated with Stream Conditions, Not Distance among Sites"

_insects, 2023, doi:10.3390/insects14110847_

Round 1
Reviewer 1 Report
Comments and Suggestions for Authors
This is an excellent paper, which should prove valuable in the design of other field studies of lotic biodiversity (e.g., what parameters are most important to cover in sampling design, where to put the greatest effort), even beyond it's examination of a specific highly important lotic system.
The Introduction (esp. lines 45-66) explains the significance and background of diversity sttudies well.
My one suggestion is to cite or define B-sor (Sorenson's index) more thoroughly for readers outside this area, near line 84 (e.g., as the authors did witth Whittaker's, line 62-64.
Author Response
Comment: My one suggestion is to cite or define B-sor (Sorenson's index) more thoroughly for readers outside this area, near line 84 (e.g., as the authors did witth Whittaker's, line 62-64.
Response: Please note we have expanded this section to now read as:
We calculated β-diversity as Sørensen pair-wise indices, βSor, among pairs of stream sites [28]. βSor is calculated using presence/absence data and can be expressed as:
βSor =2C / S1 +S2
Where C is the number of species the two communities have in common, S1 is the total number of species found in community 1, and S2 is the total number of species found in community 2 [11]. Hence this index can be viewed as the dissimilarity in species composition between any two sites.
Accordingly our aim was to explore how patterns of βSor of lotic insects, in these historically understudied coastal streams of Alabama, changed over gradients of environmental conditions and distance. We also explored how season (fall, summer) and different taxonomic resolution (genus only, genus/species mix, species only) influenced patterns of βSor. We then extended the use of the Sørensen pair-wise index to a multiple-site dissimilarity index, βMult. [49]. Whereas Sørensen is an index of two sites in a region, βMult provides a measurement of dissimilarity among all sites in a region. This latter index can further divided into diversity due to species turnover (change) among sites (βTurn) and that due to species nestedness (loss) among sites (βTurn). Hence βMult = (βTurn) + (βNest).
Reviewer 2 Report
Comments and Suggestions for Authors
In this paper the authors investigate beta diversity of stream insects in 23 sandy streams of the Mobile River system, Alabama. Beta diversity was examined in relation to distance and instream variables in what is essentially a statistical paper. A valuable aspect of the paper is the separate examination of these relationships for species, genera and mixed generic/species level identifications, and the finding that taxonomic level had no significant effect on conclusions drawn. Habitat variables had much stronger correlations with diversity than distances apart of streams, but the authors need to make it clear that distances were not great, maximum north-south and east-west dimensions of the study area each being only about 50 km. Despite the paper focusing on aquatic insects, over 20 000 of which were identified the only mention of insect taxa are the orders listed in the Methods. Many readers will have little idea of what species and genera live in this area and no indication is given as to which were numerically dominant, widespread or rare. I would like to see a table or additional text addressing these issues in broad summary and the data included in the paper as a Supplementary table.
The paper is well written and easy to follow. The Methods used are appropriate and have been used in a number of previous studies by the senior author and co-authors. Field methods are not given in detail but are summarized and the reader is referred to a previous paper. This seems reasonable. The Simple Summary, Abstract and Introduction are informative and well written and include a helpful statement on the types of diversity. The Discussion is relatively brief and could be enhanced by further mention of specific taxa.
Specific comments
Abstract Line 30 and elsewhere. Sorensen’s Index is misspelt. It is Sorensen not Sorenson.
Line 35. “correlated with”
Line 74. As I recall, Benke, Wallace and others worked on invertebrates in sandy streams of the southern US. These may have been in Georgia rather than Alabama but nevertheless would be worth a mention.
Figure 1. The map seems rather limited but does show the proximity of streams to one another. I don’t see why County lines are shown yet the main Mobile River is not labelled. Could the streams on which the study sites were located be drawn in?
Line 128. Leuctra not Lectura. As a genus it should be in italics.
Line 142. Reorderings (plural)
Line 148. Insert ‘site” after each.
Line 162. Should Index be inserted after dissimilarity?
Line 183. “... (Table 1) were between ... taxa numbers being higher than ... taxa.
Line 197. What aspects of riparian vegetation were considered? Presence/absence, vegetation type, cover?
Table 2. The table is fine as it stands but many ecologists will be interested in the ranges of variables found within the study area, as they will likely affect interpretation. Thus, what were the ranges of depths, velocities etc.? For example, does pH range widely, or is the correlation based on sites with a very narrow range of pHs? I suggest an additional table be added with this information.
Line 217. “Based on the correlation analyses of stream variables...” Wording.
Line 220. Mantel test correlations (not tests)
Line 246. Comparison not comparisons.
In the first paragraph of the discussion, comment on the limited size of the study area can be included (see above), and / or include it around line 280.
Line 320. Assisting with the field collections during one or both sampling seasons? I wouldn’t normally comment on acknowledgments, but this final sentence could be clearer.
Comments on the Quality of English LanguageThis is a clearly written paper of an ecological subject of contemporary interest and makes a useful addition to the field. In particular, a valuable aspect of the paper is the separate examination of these relationships for species, genera and mixed generic/species level identifications, and the finding that taxonomic level had no significant effect on conclusions drawn. The senior author has used the same methods successfully in other studies and the methods seem appropriate. As it stands this is a statistical paper and I would like to see more information provided on the insect fauna on which it is based (see my comments to authors). This would seem particularly appropriate for acceptance in a journal titled Insects (!); otherwise, it might be more suitable for Diversity.
Round 2
Reviewer 2 Report
Comments and Suggestions for Authors
The authors have revised their manuscript in accordance with the referees' comments. It is now much improved in my opinion with the addition of information on the fauna and the inclusion of two supplementary tables providing stream variable data and a faunal list. The map (Figure 1) is much better now and much more useful.
The following minor points need attention.
Line 201. Cheumatopsychidae should read Cheumatopsyche.
Table 1. The first word of the table heading should be Families. The family name Elmidae needs correcting in a number of places (not Elimidae).
New reference 28 (Sorensen 1948). The journal title and page numbers should be given in full. Biologiske Skrifter 5: 1-34.